# Association of Iron Status and Intake During Pregnancy with Neuropsychological Outcomes in Children Aged 7 Years: The Prospective Birth Cohort Infancia y Medio Ambiente (INMA) Study

**DOI:** 10.3390/nu11122999

**Published:** 2019-12-07

**Authors:** Victoria Arija, Carmen Hernández-Martínez, Mónica Tous, Josefa Canals, Mónica Guxens, Silvia Fernández-Barrés, Jesús Ibarluzea, Izaro Babarro, Raquel Soler-Blasco, Sabrina Llop, Jesús Vioque, Jordi Sunyer, Jordi Julvez

**Affiliations:** 1Nutrition and Public Health Unit, Research Group on Nutrition and Mental Health (NUTRISAM), Faculty of Medicine and Health Science, Universitat Rovira i Virgili, 43201 Reus, Spain; victoria.arija@urv.cat (V.A.); carmen.hernandez@urv.cat (C.H.-M.); monica.tous@urv.cat (M.T.); josefa.canals@urv.cat (J.C.); 2Pere Virgili Institute for Health Research (IISPV), Universitat Rovira i Virgili, 43003 Tarragona, Spain; 3Department of Psychology, Research Center for Behavioral Assessment (CRAMC), Universitat Rovira i Virgili, 43003 Tarragona, Spain; 4ISGlobal- Instituto de Salud Global de Barcelona, 08036 Barcelona, Spain; monica.guxens@isglobal.org (M.G.); silvia.fernandez@isglobal.org (S.F.-B.); jordi.sunyer@isglobal.org (J.S.); 5Biomedical Research Centre Network for Epidemiology and Public Health (CIBERESP), 28029 Madrid, Spain; mambien3-san@euskadi.eus (J.I.); vioque@umh.es (J.V.); 6Centro de Investigación Biomédica en Red de Fisiopatología de la Obesidad y Nutrición (CIBEROBN), 15706 Santiago de Compostela, Spain; 7Department of Child and Adolescent Psychiatry/Psychology, Erasmus University Medical Centre-Sophia Children’s Hospital, 3000CD Rotterdam, The Netherlands; 8Department of Health, Public Health Division of Gipuzkoa, 20014 San Sebastian, Spain; izaro.babarro@ehu.eus; 9BIODONOSTIA Health Research Institute, 20014 San Sebastian, Spain; 10Faculty of Psychology, University of the Basque Country (UPV/EHU), 20018 San Sebastian, Spain; 11Epidemiology and Environmental Health Joint Research Unit, FISABIO−Universitat Jaume I−Universitat de València, 46010 Valencia, Spain; raquel.Soler@uv.es (R.S.-B.); llop_sab@gva.es (S.L.); 12Unit of Nutritional Epidemiology, Universidad Miguel Hernandez, 03550 Alicante, Spain

**Keywords:** iron status, iron intake, pregnancy, neuropsychological function, working memory, attention, executive function

## Abstract

Early iron status plays an important role in prenatal neurodevelopment. Iron deficiency and high iron status have been related to alterations in child cognitive development; however, there are no data about iron intake during pregnancy with other environmental factors in relation to long term cognitive functioning of children. The aim of this study is to assess the relationship between maternal iron status and iron intake during pregnancy and child neuropsychological outcomes at 7 years of age. We used data from the INMA Cohort population-based study. Iron status during pregnancy was assessed according to serum ferritin levels, and iron intake was assessed with food frequency questionnaires. Working memory, attention, and executive function were assessed in children at 7 years old with the N-Back task, Attention Network Task, and the Trail Making Test, respectively. The results show that, after controlling for potential confounders, normal maternal serum ferritin levels (from 12 mg/L to 60 mg/L) and iron intake (from 14.5 mg/day to 30.0 mg/day), respectively, were related to better scores in working memory and executive functioning in offspring. Since these functions have been associated with better academic performance and adaptation to the environment, maintaining a good state of maternal iron from the beginning of pregnancy could be a valuable strategy for the community.

## 1. Introduction

Brain undergoes developmental changes throughout the different stages of the prenatal period, infancy, childhood, and adolescence. Epidemiological studies have demonstrated that optimizing brain development in early life has been associated with long-term consequences for individuals, for instance, academic performance, the level of education attained [1,2,3], and mental well-being in adulthood [4]. In particular, early-life environmental factors such as smoking, healthy lifestyle, and other factors have been suggested to play a major role in neurodevelopmental processes [5,6,7,8,9,10,11]. Accordingly, some previous studies have evidenced a relationship between suboptimal maternal diet in pregnancy and poorer mental and psychomotor development in offspring [5,6,12,13,14,15,16]. More specifically, several studies have reported that iron deficiency (ID) in pregnancy may adversely influence brain development and be associated with infant and child behavioral problems and poorer cognitive development [5,6,13,17,18,19,20,21]. More recently, it has been suggested that ID during pregnancy may not only have effects in early life [5,6,21,22,23,24] but could also be related to long-term alterations in cognitive, behavior, and social development and poorer school performance during childhood and adolescence [3,12,25,26,27]. Even so, a review concluded that the influence of maternal iron status during pregnancy on the cognitive function of offspring is inconclusive [13,28,29]. Some reasons for these inconsistencies may be related to methodological issues such as different cognitive assessments or to the time when iron status was assessed during pregnancy. Iron plays a crucial role in neurodevelopment processes (neuronal growth, myelinization, and neurotransmitter synthesis) that take place mainly in the third trimester of pregnancy, and few studies have assessed the iron status at that time [28].

However, there is also concern about the increased risk of adverse outcomes in the offspring of pregnant women with higher iron status. For example, high hemoglobin or high serum ferritin (SF) levels in the first and second trimester of gestation have been associated with poorer verbal scores and a lower full Intelligence Quotient (IQ) scale in both children and adults [27,30]. Furthermore, it has been shown that iron supplementation given to infants prenatally exposed to high hemoglobin levels (>12.8 g/dL) could be related to lower scores in IQ, spatial memory, and visual-motor integration in 10-year-old children [31]. In addition, the risk of excess iron in pregnant women at the end of pregnancy has been related to abnormal neonatal behavior (poorer state regulation and alertness), characteristics also observed in neonates of ID mothers [32].

Although some previous studies have shown that prenatal iron supplementation could improve a child’s neuropsychological outcomes [33,34], there is still much controversy and most of the evidence has examined the relationship in small populations, in developing countries, and in the short-term in the first months of life [35,36,37]. A recent meta-analysis concluded that the benefit of routine antenatal iron supplementation for neurodevelopment in offspring was not significant [29]. Moreover, cognitive development can be influenced by multiple factors that must be taken into account, such as obstetrical outcomes, infant breastfeeding, the mother’s alcohol and cigarette intake, the mother’s intellectual capacity, and the family environment [28,35,36,37]. Furthermore, to date, no prior studies have examined the effect of different iron status levels and intake during pregnancy together with other environmental factors on the child’s complex cognitive domains in the long term.

Therefore, the aim of the present study is to assess the relationship between iron status and iron intake during pregnancy and neuropsychological domains (attention, working memory, and executive functions) at age 7 years and also consider in the analysis important covariates and confounders uch as socio-demographical variables and other pre and peri-natal factors.

## 2. Materials and Methods 

### 2.1. Study Design and Population

The Infancia y Medio Ambiente study (INMA; http://www.proyectoinma.org) is a multicenter mother-child cohort study, which was designed to research the relationship of environmental exposures and diet during pregnancy to fetal and child development in different geographic areas of Spain (http://www.proyectoinma.org). Our results are based on data collected in three sub-cohorts established between 2003 and 2008 in the regions of Valencia, Sabadell (Catalonia), and Gipuzkoa (Basque Country). Details on recruitment strategies, general characteristics of the cohort samples and study design are described elsewhere [38,39].

Informed consent was obtained from all participants in each phase, and the study was approved by the hospital ethics committees in the participating regions.

### 2.2. Assessment of Iron Status and Iron Intake During Pregnancy

Iron status was assessed according to serum ferritin (SF) levels. SF was determined in the first trimester of pregnancy (first period) (median = 12.71 weeks; Inter Quartile Range = 5.85). Serum was aliquoted in 2.0 mL cryotube vials and stored at −80 °C. SF levels were measured by fluoroimmunoassay (Gipuzkoa and Sabadell region) and by immunoturbidimetry (Valencia region). Hemoglobin levels were also obtained by the first and second periods of pregnancy from medical records.

Iron intake during pregnancy was assessed with a semi-quantitative food frequency questionnaire (FFQ) including 101-items collected in the second period of gestation (to estimate dietary intakes from the fourth to the seventh months of gestation), previously developed and validated with pregnant women participating in the INMA cohort in Valencia [40]. From these questionnaires, we obtained the foods consumed in servings and we converted them to grams (g) using standard units and serving sizes specified for each food item. The iron content of food was obtained from the U.S. Department of Agriculture (USDA) and Spanish food composition tables [41,42]. We calculated iron intake by multiplying the frequency of consumption for each food item by the iron content of the portion size specified in the FFQ. The brand, use, and dose of specific iron supplements or multivitamin/mineral preparations containing iron were determined using a structured questionnaire at the same time-points as the FFQs. Daily iron intake (in mg) was calculated from the supplements. The total daily iron intake was obtained by adding the daily iron intake (in mg) from the supplements and from the diet.

We defined anemia as hemoglobin (Hb) levels <110 g/L, iron deficiency (ID) as SF < 12 μg/L, iron deficiency anemia (IDA) as SF < 12 μg/L and Hb < 110 g/L, and hemoconcentration (second period of pregnancy) as Hb > 130 g/L.

We classified the population into three groups according to SF levels: depleted iron stores (SF < 12 μg/L), normal iron stores (SF = 12–60 μg/L), and sufficient iron stores to complete pregnancy without iron supplement use (SF ≥ 60 μg/L) [43]. We obtained tertiles of the maternal total daily iron intake in the second period of pregnancy: Tertile 1 (T1) corresponded from 0.0 to 14.5 mg/day, tertile 2 (T2) from 14.5 to 30.0 mg/day, and tertile 3 (T3) >30.0 mg/day.

### 2.3. Assessment of Neuropsychological Outcomes

The N-Back Task [44,45] was used to assess working memory. This is a continuous performance working memory task that can be used for children and adults, in which the subject is required to respond whenever a stimulus (colors, letters or numbers) is shown on the screen that matches the one shown in previous trials (0-back, 1-back, or 2-back). This task is used to determine detectability (d’) (d´—capacity to distinguish signal from noise) for each condition. High d’ scores indicate better performance.

The Attentional Network Task (ANT) [45,46] was used to assess attention. A row of five yellow fish appearing either above or below a fixation point is shown. Children are asked to “feed” the central fish as quickly as possible by pressing either the right or left button on the mouse depending on the direction in which the fish in the middle is pointing while ignoring the flanking fish, which point either in the same (congruent) or opposite (incongruent) direction as the middle fish. The target is preceded by visual signals that notify either the upcoming target only (alerting cue), or the upcoming target as well as its location (orienting cue). The number of omission errors in the ANT is a measure of selective attention. The standard error of the Hit Reaction Time (HRT-SE) was also analyzed as a measure of attention variability. Higher scores in both outcomes indicate less efficiency.

The Trail Making Test (TMT) was used to assess mental flexibility and inhibitory control [46,47]. It consists of two parts: TMT-A requires the child to draw lines sequentially connecting 25 framed numbers distributed on the screen as quickly as possible. In TMT-B the child must alternate framed and encircled numbers. The response time can be obtained with this test and higher scores indicate less efficiency.

### 2.4. Potential Confounding Variables

Both mother and child characteristics were obtained from face to face interviews and from medical records and were included as potential confounding factors.

The mother’s characteristics were the following variables: cohort (Guipuzkoa, Valencia or Sabadell); age (years); social class (occupation during pregnancy based on the highest social class by using a widely used Spanish adaptation of the International ISCO88 coding system of high status I–II comprising managers/technicians, medium status III comprising skilled, low status IV–V comprising semiskilled/unskilled); educational level (primary, secondary or university); marital status at the time of neuropsychological assessment (lives with the father of the child: yes, no); parity (primiparous, multiparous); ethnic group (Caucasian, others); mode of delivery (eutocic, dystocic); smoking during pregnancy (in the first period and in the second period: yes, no); Body Mass Index (BMI, kg/m^2^); and maternal verbal IQ proxy (Wechsler Adult Intelligence Scale III, Similarity subtest scalar score).

Furthermore, adherence to the MD (Mediterranean Diet) was assessed with the relative MD score (rMED) [48]. The rMED was constructed with the average dietary data from the first and third trimesters of pregnancy. The consumption of vegetables, fruits and nuts, cereals, legumes, fish, olive oil, total meat, and dairy products (except alcohol because most of the pregnant women did not consume it in our study) were measured as grams per 1000 kcal/day and were divided into tertiles. A value of 0, 1, and 2 were assigned to the intake tertiles, positively scoring higher intakes for the six components presumed to fit the MD. The scoring was reversed for two components presumed not to fit the MD (meat and dairy), positively scoring lower intakes. After summing up the scores of each component, the final potential score range was 0–16; it was further divided into tertiles to identify those with low, medium, and high adherence to the MD.

C-reactive protein (CRP) levels were also measured in the first period of pregnancy by using an immunoturbidimetric method at Consulting Quimico Sanitario (CQS) Laboratory (Gipuzkoa region), and the Echevarne Sabadell Laboratory for the Sabadell region. CRP levels were not measured for the Valencia region. CRP levels were used to assess inflammation (defined as CRP levels ≥10 mg/L) and to complement SF information. In the samples in which CRP levels were measured, none of the pregnant women had values above 10 mg/L.

We measured organochlorine compound (OC) concentrations in maternal serum samples of the three cohorts (Sabadell, Gipuzkoa, and Valencia). Samples collected in Gipuzkoa and Sabadell were analyzed at the Basque government’s Public Health Laboratory in San Sebastian (limit of detection (LOD) of 0.071 ng/mL for all the OCs), and samples from Valencia were analyzed at the Barcelona Institute of Environmental Assessment and Water Research (LODs of OCs between 0.010 and 0.035 ng/mL), as previously described [49,50]. OC concentrations were determined by gas chromatography with electron capture detection and confirmation by gas chromatography coupled to a mass spectrometer detector. Both laboratories complied with the Arctic Monitoring and Assessment Program for persistent organic pollutants in human serum (Centre de Toxicologie, Institut National de Sante Publique du Quebec). In this study, we present the results of those OCs with a detection frequency >80%: 4,4´-dichlorodiphenyldichloroethylene (DDE), and polychlorinated biphenyls 138, 153, and 180 (PCBs). Values below the LOD were replaced by values within the range (0, LOD) based on a multiple imputation procedure. We classify the values of the PCBs in smaller and greater than 110.9 ng/g of lipids, and DDE levels in smaller and greater than 123.0 ng/g of lipids.

The child’s characteristics were the following variables: standard weight at gestational week 40 (grams), standard height at gestational week 40 (cm), breastfeeding (months), the child’s age at the moment of neuropsychological examination (years), and child’s gender (male, female).

### 2.5. Statistical Analysis

Descriptive analyses of the general characteristics of mothers and children were performed and differences according to geographical area/iron supplement use/total iron intake were assessed using the chi-squared test for categorical variables and *t*-Test/ANOVA for continuous parametric variables or Mann-Whitney U test/Kruskal-Wallis test for non-parametric variables.

Differences in child neuropsychological outcomes between the three groups defined by maternal SF levels or tertiles of total iron intake were identified using the Kruskal–Wallis test. Normality analyses were performed using the Shapiro Wilk test.

Multiple linear regression models using the stepwise method were performed for each neuropsychological outcome assessed and taking into account separately maternal SF levels in the first trimester of pregnancy (SF < 12 μg/L; SF from 12 to 60 μg/L and SF ≥ 60μg/L) and total iron intake (<14.5 mg/day; 14.5–30.0 mg/day; >30 mg/day) in the second period of pregnancy. The candidate variables entered in the models were the potential confounding variables described above.

To control the increase in type I error, we applied the Bonferroni correction and the statistical significance was determined at *p* < 0.02. Statistical analyses were performed using SPSS for Windows 25.0 (SPSS Science, Chicago, IL, USA).

## 3. Results

### 3.1. Descriptive Results

The general characteristics of pregnant women in relation to the geographical area are presented in Table 1. In general, the groups differed in socio-demographic characteristics, maternal IQ, smoking and alcohol use.

Table 2 shows the sample characteristics according to maternal serum ferritin (SF) levels and total iron intake. In relation to the maternal levels of SF measured in the first period, we observed that there are no pregnant woman with ID or IDA in the SF group >12 mg/L, as it was expected. In addition, women of this group had higher tobacco use and alcoholic beverages consumption, higher levels of PCR, of DDE, and PCBs.

In relation to the total maternal intake of iron in the second gestation period (including iron supplements), we observed that lower levels of maternal education and social class were related with higher intake. The increasing iron intake was associated with slightly better levels of SF and lower percentages of anemia. In addition, a correlation was observed with higher levels of PCR, of DDE, and PCBs.

### 3.2. SF Levels and Neuropsychological Outcomes in Children

Table 3 shows SF levels measured in the first period of pregnancy (first trimester of pregnancy) and neuropsychological outcomes in children at age 7–8 years. With the N-Back Test, we observed that offspring of mothers who had SF levels from 12 to 60 μg/L during pregnancy had higher scores in the d’ numbers 2-back N-Back test than offspring of mothers that had SF levels <12.0 μg/L. A similar pattern was observed with TMT A scores.

However, although trends can be observed, after applying the Bonferroni correction no significant differences were obtained in the N-Back Test, Attention Network Test or Trail Making Test between the different groups of SF levels.

### 3.3. Total Iron Intake and Neuropsychological Outcomes in Children

Table 4 shows the maternal iron intake in the second period of pregnancy and neuropsychological outcomes of children assessed at 7–8 years of age.

Higher d’ numbers 2 n-back scores were observed, comparing T2 to T1 tertiles of total daily iron intake (*p* = 0.014). In the ANT, higher omission errors, indicating less efficiency of networks, were observed in the group of children whose mothers had the highest daily iron intake during pregnancy, whereas no changes were observed in the HRT-SE by tertiles of iron intake. In addition, significantly higher TMT A scores, indicating a lower speed of processing, were observed in children whose mothers had a daily iron intake > 14.50 mg/day, compared to those with an intake <14.50 mg/day.

### 3.4. Association of Child Executive Functioning with SF Concentrations in the First Period of Pregnancy, and with Total Iron Intake in the Second Period of Pregnancy

Multiple linear regression models were performed to assess the predictive capacity of SF levels and total iron intake for child neuropsychological outcomes at 7–8 years of age. Significant models are shown in Table 5. Significant associations were detected between maternal SF levels and total iron intake and working memory (N-Back task) assessed in the child. Specifically, children of mothers with SF levels 12–60 μg/L and >60 µg/L in the first period of pregnancy, performed better in the N-Back task and in particular, in the d’ numbers 2 backload (B = 0.38; *p* = 0.011 and B = 0.46; *p* = 0.013 respectively). In addition, the group of children whose mothers had normal maternal SF levels during gestation had faster responses in the Trail Making Test A (B = −7.91; *p* = 0.017). In relation to total iron intake in the second period of pregnancy, children of mothers in the second tertile and third tertile obtained higher scores in the d’ numbers 2 back load (B = 0.30; *p* = 0.014 and B = 0.30; *p* = 0.016), and children of mothers in the third tertile obtained higher scores in the d’ numbers 3 back (B = 0.23; *p* = 0.024).

Linear regression analyses with the stepwise method. Neuropsychological outcomes in 7-year-old children as dependent variable. Independent variables: SF levels and tertile categories of total iron intake (T1: 0.0–14.5 mg/day; T2: 14.5–30 mg/day; T3: >30 mg/day). Covariate variables entered in the models: Hemoglobin levels, SF (in models where the independent variable is total iron intake), cohort (Guipuzkoa, Valencia or Sabadell); child age (years); social class (high; medium; low); mother’s educational level (primary, secondary or university); marital status at the time of neuropsychological assessment (lives with the child’s father: no, yes); parity (primiparous, multiparous); ethnic group (Caucasian, others); mode of delivery (vaginal or instrumental); smoking during pregnancy (no, yes); BMI (continuous); standard weight at gestational week 40 (grams); maternal IQ approximation (WAIS-III Similarity scalar score); child’s gender (male, female); breastfeeding (months) and child’s age at the time of neuropsychological examination (years); maternal Mediterranean diet score during first 12 weeks of pregnancy; maternal DDE and PCBs concentrations during 12 weeks of pregnancy.

## 4. Discussion

In this study, we assessed the association of both maternal iron status and total iron intake during pregnancy with cognitive functioning of children aged 7 years from the Spanish INMA population-based birth cohort study. In this cohort, we found that maternal SF levels from 12 to 60 μg/L in the first trimester of pregnancy were associated with significantly higher scores in the neuropsychological outcomes, such as working memory and executive function compared with children of mothers with SF levels <12 μg/L. Similarly, the children of the group of mothers with levels of SF > 60 μg/L also had better results in working memory tasks. In addition, after adjusting for potential confounding factors, we found that a total iron intake higher than 14.50 mg/day from the second period of pregnancy was indeed associated with significantly better performance in working memory tasks by their children.

To assess the iron status of pregnant women, we measured SF levels and hemoglobin levels in the first period of pregnancy. In particular, we found that 14.0% of pregnant women in the first period showed ID (SF < 12 µg/L), whereas the prevalence of IDA was very low (ranging from 0 to 0.6%). Our results are in agreement with previous studies that have estimated the prevalence of ID and IDA in pregnant women in European countries [51,52,53]. According to previous studies, in our sample, the prevalence of anemia had a clear tendency to increase during pregnancy [43], going from 2.2% to 22.7%. In our study, almost half of the women (46.4%) had normal iron stores (SF > 30 μg/mL) and 13.4% had levels >60 μg/mL, and thus, were considered to have sufficient iron stores to undergo pregnancy without supplementation [54].

Overall, our findings are consistent with those of previous studies demonstrating the potential importance of maternal iron status in fetal programming and lifelong health [3,4,5,6,13,20,25,55,56,57]. Although a previous review concluded that maternal nutritional status has no effect on offspring neurodevelopmental outcomes, [28] differences in iron status indicators, iron measurements during pregnancy, and the domains assessed [58] could explain some of these inconsistencies. The results of our study also show that in addition to initial iron status, reflected by SF levels, a total intake of iron higher than 14.50 mg/day was associated with higher working memory scores in their children at 7 years, a key executive function that influences other more complex executive functions. Although the differences found in these functions between groups are not clinically relevant, it is important to highlight their effect as it is found in a well-nourished community sample from a developed country. On average, women were supplemented with iron doses of 17.67 mg/day from the fourth to seventh month of pregnancy and our data, in accordance with previous studies, suggest that this dose of iron could be enough to improve cognitive functioning in children of pregnant women who are at risk of obtaining insufficient amounts of iron from their diet. However, although the WHO has proposed international iron supplementation recommendations, to date there is no international consensus between countries and scientific societies, and iron recommendations in pregnancy vary widely across developed countries [54]. Epidemiological studies evidence that iron supplements during pregnancy are associated with improved maternal iron status [52] and according to previous findings, they can prevent adverse birth outcomes in their offspring [3,5,21,23,25,26,27,59]. However, some previous studies have reported a high prevalence of ID and IDA among pregnant women even when they were given iron supplements during pregnancy [60], probably because the supplementation doses were insufficient. In particular, we found that 51.9% of ID women were not taking iron supplements in the first period of pregnancy and 36.3% of ID women showed a total iron intake lower than 14.50 mg, which identifies a large group of pregnant women who, despite having iron deficiency, perform a risk behavior by not taking iron supplements and having a low iron intake, which can have future consequences for their offspring’s cognitive functioning. Previous studies with animal models suggest that early-life ID could have long-lasting effects on cognitive function independently of receiving an iron-normal diet later in life [61]. The small sample size in this subgroup of women makes it difficult to confirm this interpretation [62].

Taken together, our results are in line with previous findings and suggest that it is important to consider SF levels at the beginning of pregnancy to identify women who are at risk of ID, and thus improve the cognitive performance of their offspring.

Previous evidence suggests a higher risk for poorer neurodevelopmental outcomes in children born to mothers with SF levels >60 ug/L in the first and second trimester of gestation [27,31], exhibiting a U-shaped risk, however, in our study we have not observed this association. Specifically, we did not find substantial differences in neuropsychological outcomes in children whose mothers had SF levels from 12 to 60 μg/L and those with SF levels ≥60 μg/L.

Further studies are necessary that include randomized controlled trials to examine in depth the neuropsychological outcomes of offspring in relation to maternal iron status in early pregnancy. Different supplement doses are necessary to understand the impact of both maternal iron status and total iron intake on neurodevelopmental processes. 

To date, the mechanisms by which ID could be related to poorer neuropsychological outcomes in children in the long term are not clear. Although a number of studies have assessed the mechanisms by which higher iron status during pregnancy could be related to adverse ID outcomes in offspring, they are inconclusive and most of the studies that reported cognitive functioning in children were performed with children less than 35 months in age. Early life ID could result in adverse effects on myelination [23,63], neurotransmitter synthesis [64,65], and hippocampus development [66]. It has been shown that ID could indeed alter recognition memory [23,26,59] and lead to lasting impairments in recall memory [63], functions in which the hippocampus itself appears to be critically involved [67,68]. Accordingly, recent studies performed with experimental animals indicate that reduced iron content and changes in DNA methylation in the hippocampus followed early-life ID [69,70]. Although previous work has evidenced that the development of executive functions mainly depends on the prefrontal cortex, it also depends on the maturation of other regions and existing connections between these and the prefrontal cortex, including the connections with subcortical regions (mainly the basal ganglia, thalamus, and hippocampus) [71].

It is important to note that the effect of nutrients on brain function appears to occur both in the short- and long-term once maturation has been reached. For instance, the prefrontal cortex develops during the second year of life, between 7 and 9 years of age and at 15 years of life [72]. Other factors, such as mode of delivery, breastfeeding, Mediterranean diet, maternal IQ, SES, marital status (covariates in the analysis) and other factors like psychosocial stimulation, mental health of caregivers, child’s nutrition, sleep and physical activity, among others, could indeed affect brain development and cognitive function and consequently minimize the detrimental effects on the child’s mental health of nutritional deficits during the fetal period. For instance, strong correlations have been found between the cognitive abilities of parents and their children, which is stronger in higher socioeconomic contexts [73]. We have considered the mothers’ Mediterranean diet as adjustment variables in this study to be able to control the effect on the bioavailability of iron that has various components of the diet (protein factor, vitamin C, calcium, etc.) [54], and also to control the possible effect that other maternal nutrients, such iron, have on the neurodevelopment of their child [1,16,28,33]. However, no differences were observed in the Mediterranean diet score between the tertiles of maternal FS or maternal iron intake, indicating that the pattern of the diet carried out by pregnant women did not significantly influence the neurodevelopment of the child, as also verified in the Logistic Regressions carried out. Furthermore, a well-balanced microbial community in early life has been reported to play a major role in neurodevelopmental outcomes throughout the lifespan [74]. Hence, we cannot exclude the possibility that the above factors probably also impact the child’s cognitive abilities assessed at 7 years.

Although previous studies have linked ID during pregnancy to neurodevelopmental disorders [75,76,77], we did not study the association of ID with the risk of neurodevelopment disorders in offspring. However, the neuropsychological outcomes assessed, working memory, flexibility, attention, and impulsivity are all altered in certain neurodevelopmental disorders, such as Attention Deficit Hyperactivity Disorder (ADHD), the most common childhood neuropsychiatric disorder characterized by hyperactivity, impulsivity, and/or lack of attention. The research into the effect of both iron status and iron intake in pregnancy on the risk of neurodevelopmental disorders might be of interest, particularly taking into account the effect of important covariates and confounders.

Before finishing, it is important to discuss the strengths and limitations of the study. In the present study, we have assessed the association of both maternal iron status and iron intake during pregnancy with executive functioning of a sample of 7-year old children from the INMA population-based birth cohort study with a sample size large enough to obtain adequate statistical power. Regarding the main variables, iron status during pregnancy was assessed with two indicators: SF levels, the most clinically applicable biomarker used to define ID in pregnancy; and hemoglobin levels, which helped us interpret the results. Although the assays to obtain SF were different in the Valencia Cohort, the methods used are comparable [78]. We also measured maternal CRP levels. The results showed very low levels of this inflammation biomarker, which suggests that in our sample, SF levels >60 ug/L are not related to inflammation or inadequate volume expansion and consequently reflect normal iron stores in the body during early pregnancy. Iron intake during pregnancy (dietary and supplements) was assessed with a validated food frequency questionnaire and detailed information on dosages of iron supplements were obtained. The neuropsychological assessment, was composed of highly valid and reliable tasks that are widely used in clinical, epidemiological and neuroimaging studies. Finally, a wide range of confounding factors from the prenatal, perinatal, and postnatal periods were assessed and used in the regression analysis. However, a large limitation is that we did not measure other important variables that affect infant neuropsychological outcomes, such as the physical activity level or nutritional status of the children.

Taking all of this into account, our results allow us to conclude that both maternal iron status and iron intake during pregnancy are associated with the neuropsychological function in children. In particular, children of women with SF > 12 μg/L in the first period of pregnancy or with total iron intake higher than 14.50 mg/day showed a better executive cognitive function. Given that executive functioning has been related to better academic performance in childhood, evaluating these indicators in order to optimize cognitive status could be a valuable strategy for societies.

## Figures and Tables

**Table 1 nutrients-11-02999-t001:** General characteristics of the INMA cohort study, Spain 2003–2008 by geographical areas.

General Characteristics	All Cohorts	Guipuzkoa	Sabadell	Valencia	*p*
	(*n* = 2032)	(*n* = 618)	(*n* = 634)	(*n* = 780)	
Maternal age, years, Mean (SD) ^1^	30.33 (4.4)	31.37 (3.7)	30.23 (4.5)	29.61 (4.6)	<0.001
Maternal pre-pregnancy BMI, kg/m^2^, Mean (SD)^1^	23.52 (4.3)	23.03 (3.7)	23.71 (4.5)	23.75 (4.6)	0.003
Mode of delivery, Eutocic vs. Dystocic, *n* (%) ^2^	1171 (62.3)	394 (70.1)	395 (67.3)	382 (52.2)	<0.001
Parity ≥ 1 vs. 0, *n* (%) ^2^	909 (44.8)	287 (46.4)	280 (44.3)	342 (43.8)	0.600
Ethnic group					
Caucasian vs. other, *n* (%) ^2^	1946 (95.9)	603 (98.0)	613 (96.7)	730 (93.5)	<0.001
Maternal marital status					
Lives with father’s child vs. other situations, *n* (%) ^2^	1998 (98.3)	615 (99.5)	624 (98.4)	759 (97.3)	0.006
Maternal educational level, *n* (%) ^2^					
Primary or no education	531 (26.2)	82 (13.3)	183 (29.0)	266 (34.1)	<0.001
Secondary	830 (40.9)	227 (36.9)	267 (42.2)	336 (43.1)	
University	667 (32.9)	307 (49.8)	182 (28.8)	178 (22.8)	
Social Class, *n* (%) ^2^					
High	392 (19.8)	151 (24.8)	135 (22.6)	106 (13.7)	<0.001
Medium	338 (17.1)	80 (13.2)	107 (17.9)	151 (19.5)	
Low	1248 (63.0)	377 (62.0)	356 (59.5)	515 (66.5)	
Smoking, yes vs. no, *n* (%) ^2^					
1st period *	354 (18.4)	73 (12.5)	91 (15.3)	190 (25.5)	<0.001
2nd period *	326 (16.9)	66 (11.3)	85 (14.3)	175 (23.5)	<0.001
Alcohol intake, g/day, Mean (SD) ^3^					
1st period *	0.33 (1.2)	0.18 (0.7)	0.33 (1.3)	0.41 (1.5)	0.003
2nd period *	0.33 (1.2)	0.23 (0.7)	0.36 (1.2)	0.39 (1.3)	0.025
Breastfeeding, months, Mean (SD) ^1^	25.8 (19.8)	28.9 (20.6)	25.9 (19.2)	19.5 (19.8)	<0.001
Maternal IQ, Mean (SD) ^1^	10.1 (3.0)	9.8 (2.7)	10.6 (2.9)	9.8 (3.3)	<0.001

^1^ ANOVA for continuous parametric variables; ^2^ Chi-squared test; ^3^ Kruskal-Wallis test for non-parametric variables; * 1st period: First period of pregnancy; 2nd period: Second period of pregnancy; *n* varies across assessments; INMA, Infancia y Medio Ambiente.

**Table 2 nutrients-11-02999-t002:** General characteristics and iron status parameters of pregnant women and child’s characteristics from the INMA cohort study, Spain 2003–2008.

General Characteristics	Serum Ferritin Levels (μg/L) 1st Period of Pregnancy				Total Iron Intake (mg/day) 2nd Period of Pregnancy			
	<12.0	12.0–60.0	>60	*p*	<14.5	14.5–30.0	>30.0	*p*
Maternal age, years Mean (SD) ^1^	30.8 (4.4)	30.3 (4.4)	30.3 (4.1)	0.287	30.3 (4.1)	30.4 (4.2)	30.4 (4.5)	0.803
Maternal pre-pregnancy BMI, kg/m^2^, Mean (SD) ^1^	23.0 (3.7)	23.5 (4.4)	24.0 (4.1)	0.23	23.8 (4.4)	23.3 (4.1)	23.5 (4.3)	0.081
Mode of delivery, Eutocic vs. Dystocic, *n* (%) ^2^	170 (70.2)	807 (63.1)	128 (55.2)	0.003	400 (64.6)	376 (61.3)	376 (60.6)	0.305
Ethnic group, Caucasian vs. other, *n* (%) ^2^	252 (96.9)	1307 (96.4)	238 (98.4)	0.403	616 (96.7)	618 (96.4)	609 (95.3)	0.932
Maternal educational level, *n* (%) ^2^								
Primary or no education	63 (24.2)	347 (25.6)	75 (29.9)	0.349	140 (21.9)	167 (26.1)	179 (28.1)	0.008
Secondary	102 (39.2)	561 (41.4)	90 (35.9)		262 (40.9)	247 (38.7)	275 (43.1)	
University	95 (36.5)	446 (32.9)	86 (34.3)		238 (37.2)	225 (35.2)	184 (28.8)	
Maternal Social Class, *n* (%) ^2^								
High	56(21.5)	283 (20.8)	59 (23.5)	0.909	148 (23.1)	139 (21.7)	126 (19.7)	0.025
Medium	74 (28.4)	377 (27.7)	68 (27.1)		197 (30.8)	167 (26.1)	162 (25.3)	
Low	131 (50.2)	699 (51.1)	124 (49.4)		295 (46.1)	335 (52.3)	352 (55.0)	
Mediterranean diet, score, Mean (SD) ^1^								
1st period *	7.1 (2.4)	7.0 (2.4)	6.9 (2.4)	0.248	7.1 (2.4)	7.0 (2.4)	6.9 (2.4)	0.248
Smoking, yes vs. no, *n* (%) ^2^								
1st period *	29 (11.6)	244 (18.8)	52 (21.9)	0.007	121 (19.0)	112 (17.5)	120 (18.8)	0.757
2nd period *	26 (10.4)	224 (17.2)	49 (20.7)	0.007	107 (16.8)	106 (16.6)	113 (17.7)	0.854
Alcohol intake, g/day, Mean (SD) ^3^								
1st period *	0.2 (0.7)	0.3 (1.1)	0.5 (1.6)	0.022	0.3 (1.1)	0.3 (1.2)	0.4 (1.3)	0.532
2nd period *	0.3 (0.8)	0.3 (1.2)	0.4 (1.2)	0.372	0.3 (1.2)	0.4 (1.1)	0.4 (1.2)	0.812
**Iron Characteristics and other Parameters**								
Hemoglobin, g/L, Mean (SD) ^1^								
1st period *	128.5 (9.0)	128.9 (9.1)	129.1 (9.0)	0.760	128.2 (9.2)	129.2 (8.2)	129.2 (8.3)	0.085
2nd period *	11.8 (1.2)	11.8 (1.1)	11.8 (1.1)	0.968	11.7 (1.1)	11.9 (1.1)	11.9 (1.1)	0.022
Anemia, *n* (%) ^2^								
1st period *	4 (1.7)	37 (3.1)	4 (1.8)	0.362	26 (4.6)	10 (1.7)	8 (1.4)	0.001
2nd period *	69 (30.0)	348 (28.0)	66 (28.8)	0.824	183 (31.3)	153 (25.7)	163 (26.8)	0.078
Iron deficiency (ID), *n* (%)^2^								
1st period *	261 (100.0)	0 (0.0)	0 (0.0)	<0.001	90 (14.7)	76 (12.7)	82 (14.2)	0.590
Iron deficiency anemia (IDA), *n* (%) ^2^								
1st period *	4 (1.7)	0 (0.0)	0 (0.0)	<0.001	3 (0.5)	0 (0.0)	1 (0.2)	0.214
Iron supplementation, *n* (%) ^2^								
2nd period *	124 (48.1)	694 (53.0)	129 (53.3)	0.329	63 (10.4)	347 (56.5)	637 (99.7)	<0.001
CRP, mg/L Mean (SD) ^3^								
1st period *	0.6 (0.7)	0.6 (0.7)	0.9 (1.6)	0.003	0.6 (0.7)	0.6 (0.7)	0.6 (0.7)	0.375
PCBs, *n* (%) ^2^ <110.9 vs. >110.9 ng/g lipid								
1st period *	54.9 (139)	50.2 (646)	49.3 (113)	0.351	53.3 (327)	48.4 (288)	45.6 (264)	0.027
DDE, *n* (%)^2^ <123.0 vs. >123.0 ng/g lipid								
1st period*	54.2 (137)	51.6 (663)	54,4 (104)	0.135	53.3 (327)	53.4 (318)	43.5 (252)	0.001
**Child’s Characteristics**								
Gender of the child, Male, *n* (%) ^2^	124 (49.8)	671 (51.2)	123 (51.7)	0.901	327 (51.7)	314 (49.6)	329 (51.9)	0.669
Standard birth weight, grams, Mean (SD) ^1^	3376.7 (406.4)	3327.4 (399.8)	3327.9 (393.6)	0.203	3331.6 (381.4)	3342.2 (414.7)	3335.3 (408.0)	0.894
Standard birth height, cm, Mean (SD) ^1^	49.9 (1.7)	49.8 (1.7)	49.8 (1.9)	0.688	49.7 (1.8)	49.8 (1.8)	50.1 (1.8)	<0.001
Standard birth head circumference, cm, Mean (SD) ^1^	34.51 (1.2)	34.43 (1.3)	34.44 (1.8)	0.744	34.44 (1.2)	34.52 (1.2)	34.35 (1.2)	0.065
Breastfeeding, months, Mean (SD)^1^	28.1 (20.1)	25.6 (20.0)	27.2 (20.0)	0.152	26.2 (20.1)	26.7 (20.3)	24.9 (20.0)	0.278
Age at neuropsychological examination, years, Mean (SD)^2^	7.3 (0.6)	7.4 (0.5)	7.4 (0.5)	0.334	7.3 (0.6)	7.4 (0.5)	7.5 (0.4)	0.001

^1^ ANOVA for continuous parametric variables; ^2^ Chi-squared test; ^3^ Kruskal-Wallis test for non-parametric variables; * 1^st^ period: First period of pregnancy; 2^nd^ period: Second period of pregnancy; Standard: Standard at gestational week 40; INMA, Infancia y Medio Ambiente; *n* varies across assessments; INMA, Infancia y Medio Ambiente.

**Table 3 nutrients-11-02999-t003:** Neuropsychological scores in children aged 7–8 years according to serum ferritin (SF) levels (μg/L) measured in the first period of pregnancy. INMA cohort study, Spain 2003–2008.

	Groups of SF (μg /L)			*p*	Posthoc Analyses
	<12.0 ^a^ Median (IQR)	12.0–60.0 ^b^ Median (IQR)	≥60.0 ^c^ Median (IQR)		
**N-Back Test**					
d’ numbers 1-back	3.4 (3.3–3.6)	3.4 (3.4–3.5)	3.5 (3.4–3.6)	0.744	
d’ numbers 2-back	1.6 (1.4–1.8)	1.9 (1.8–1.9)	1.9 (1.7–2.1)	0.037	0.029 ^ab^
d’ numbers 3-back	1.0 (0.86–1.17)	1.1 (1.0–1.1)	1.1 (1.0–1.3)	0.691	
**Attention Network Test**					
Omission errors	4.9 (3.5–6.2)	4.8 (4.3–5.3)	5.0 (3.8–6.1)	0.963	
HRT-SE	317.4 (306.2–328.8)	316.0 (310.8–321.3)	316.1 (303.5–328.8)	0.979	
**Trail Making Test**					
Response time TMT A	78.8 (70.9–86.7)	71.5 (69.3–73.7)	72.1 (68.1–76.1)	0.050	0.041 ^ab^
Response time TMT B	77.9 (71.5–84.3)	73.9 (71.7–76.2)	74.0 (69.0–79.1)	0.413	

*p*-values are from Kruskal–Wallis test; IQR: Inter Quartile Range; HRT SE: Hit reaction time standard error; ^a^ Ferritin levels in the first period of pregnancy <12.0 μg/L; ^b^ Ferritin levels in the first period of pregnancy 12.0–60.0 μg/L; ^c^ Ferritin level in the first period of pregnancy >60.0 μg/L; ^a,b^ Differences between groups.

**Table 4 nutrients-11-02999-t004:** Neuropsychological scores in children aged 7–8 years according to total daily iron intake (mg/day) assessed in the second period of pregnancy. INMA cohort study, Spain 2003–2008.

	Tertiles of Total Iron Intake (mg/day)			*p*	Posthoc Analyses
	0.0–14.5 ^a^	14.5–30.0 ^b^	>30.0 ^c^		
	Median (IQR)	Median (IQR)	Median (IQR)		
**N-Back Test**					
d’ numbers 1-back	3.4 (3.3–3.5)	3.5 (3.4–3.6)	3.4 (3.3–3.5)	0.587	
d’ numbers 2-back	1.7 (1.6–1.8)	2.0 (1.8– 2.07)	1.8 (1.7–1.9)	0.045	0.014 ^ab^
d’ numbers 3-back	0.9 (0.9–1.0)	1.1 (1.0–1.2)	1.1 (1.0–1.2)	0.131	
**Attention Network Test**					
Omission errors	4.5 (3.9–5.1)	4.7 (3.9–5.6)	5.5 (4.7–6.2)	0.022	0.045 ^ab^ 0.029 ^ab^
HRT- SE	315.8 (308.8–322.8)	312.1 (304.5–319.8)	321.4 (313.3–320.8)	0.146	
**Trail Making Test**					
Response time TMT A	70.3 (67.4–73.3)	75.2 (71.6–78.9)	72.0 (67.9–76.0)	0.014	0.004 ^ab^
Response time TMT B	73.2 (70.3–76.1)	75.4 (72.0–78.7)	74.6 (69.7–79.4)	0.467	

*p*-values are from Kruskal–Wallis test; IQR: Inter Quartile Range; HRT SE: Hit reaction time standard error; ^a^ Total iron intake in the second period of pregnancy 0.0–14.5; ^b^ Total iron intake in the second period of pregnancy 14.5–30.0; ^c^ Total iron intake in the second period of pregnancy >30.0; ^a,b^ Differences between groups.

**Table 5 nutrients-11-02999-t005:** Regression models for predicting neuropsychological outcomes in children aged 7–8 years according to serum ferritin (SF) levels (mg/L) measured in the first period of pregnancy and total iron intake (mg/day) in the second period of pregnancy. INMA cohort study, Spain 2003–2008.

**N-Back Test**	**B (95%CI)**	***p***	**Trail Making Test A**	**B (95%CI)**	***p***
**d’ numbers 2- back**			**Response time in seconds**		
SF (<12 µg/L vs. 12–60 µg/L)	0.4 (0.1–0.7)	0.011	SF (<12 µg/L vs. 12–60 µg/L)	−7.9 (−14.5–−1.4)	0.017
SF (<12 µg/L vs. >60 µg/L)	0.5 (0.1–0.8)	0.013	SF (<12 µg/L vs. >60 µg/L)	−6.9 (−15.4–1.7)	0.114
**N-Back Test** **d’ numbers 2- back**			**N-Back Test** **d’ numbers 3- back**		
Total iron intake (T1 vs. T2)	0.3 (0.1–0.5)	0.014	Total iron intake (T1 vs. T2)	0.1 (−0.1–0.3)	0.189
Total iron intake (T2 vs. T3)	0.3 (0.1–0.5)	0.016	Total iron intake (T2 vs. T3)	0.2 (0.1–0.4)	0.024

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
