# Peer review of "Association of Iron Status and Intake During Pregnancy with Neuropsychological Outcomes in Children Aged 7 Years: The Prospective Birth Cohort Infancia y Medio Ambiente (INMA) Study"

_nutrients, 2019, doi:10.3390/nu11122999_

Round 1
Reviewer 1 Report
The manuscript is markedly improved from previous submissions; I have suggested a few adjustments and will point out some minor errors that I believe the authors will be able to address.
My only two "major" points are:
In table 1 and 2 the authors state the various statistical analysis methods used and write the type of variable and the method used - but this is not indicated in the actual table itself. It should not be up to the reader to have to identify the variable/data type to figure out what analysis was used - this should be clearly indicated in the tables.
The authors have included analysis of Mediterranean Diet - but I see no discussion of this - did it matter? This should be clarified - and also why they felt that in a study of iron deficiency and pregnanacy Med diet is important to analyse.
Other points are as follows:
line 48 - epidemiological studies demonstrate association - please reflect in the sentence
line 81- what is meant by "toxic habits" - also perjhaps the authors would consider using a less judgemental word?
lines 102 and 103 - please check grammer/punctuation
lines 104-105 - have the two types of assays been validated against each other?
line 124 - please check grammar
line 168 - please check grammar
lines 178-179 - CRP findings belong in results not methods
line 222- suggest maternal rather than mother
line 224 - what does different mean?
line 230 -please check grammar
Tables 1& 2 as discussed
lines 259-270 - I believe this is where Med diet was analysed - what were the findings? And why was it included?
line 306 - evidence to support this - do Haemoglobin levels in WHO for pregnant women not already reflect expanded plasma volume (they are lower than levels for non-pregnant women)
line 331- evidence to support this statement?
line 335 - please check grammar
line 343 - I am not sure what is meant by the statement beginning " any adverse.....
Author Response
Comments and Suggestions for Authors
The manuscript is markedly improved from previous submissions; I have suggested a few adjustments and will point out some minor errors that I believe the authors will be able to address.
My only two "major" points are:
In table 1 and 2 the authors state the various statistical analysis methods used and write the type of variable and the method used - but this is not indicated in the actual table itself. It should not be up to the reader to have to identify the variable/data type to figure out what analysis was used - this should be clearly indicated in the tables.
AUTHORS: We have clarified what analysis have been made in each variable.
The authors have included analysis of Mediterranean Diet - but I see no discussion of this - did it matter? This should be clarified - and also why they felt that in a study of iron deficiency and pregnanacy Med diet is important to analyse.
AUTHORS: We have added a text about the need of inclusion the Mediterranian Diet in the analysis in the discussion section.
Other points are as follows:
line 48 - epidemiological studies demonstrate association - please reflect in the sentence
AUTHORS: We have reflected it in the sentence.
line 81- what is meant by "toxic habits" - also perjhaps the authors would consider using a less judgemental word?
AUTHORS: We meant mainly alcohol and cigarettes intake. We have specified this in the sentence.
lines 102 and 103 - please check grammer/punctuation.
AUTHORS: Done.
lines 104-105 - have the two types of assays been validated against each other?
AUTHORS: Yes, these two assays are comparable and we had a sentence in the limitations paragraph (lines 387-388). “Although the assays to obtain SF were different in the Valencia Cohort, the methods used are comparable [78].”
line 124 - please check grammar
AUTHORS: Done.
line 168 - please check grammar
AUTHORS: Done.
lines 178-179 - CRP findings belong in results not methods
line 222- suggest maternal rather than mother
AUTHORS: Done.
line 224 - what does different mean?
AUTHORS: The sentence has been corrected.
line 230 -please check grammar
AUHTORS: Done.
Tables 1& 2 as discussed
lines 259-270 - I believe this is where Med diet was analysed - what were the findings? And why was it included?
AUHTORS: We have included in discussion section the following sentence: “We have considered the mothers' Mediterranean diet as adjustment variables in this study to be able to control the effect on the bioavailability of iron that have various components of the diet (protein factor, vitamin C, calcium, etc.) [55], and also to control the possible effect that other maternal nutrients, further iron, have on the neurodevelopment of their child [1,16, 29, 34]. However, no differences were observed in the Mediterranean diet score between the tertiles of maternal FS or maternal iron intake, indicating that the pattern of the diet carried out by pregnant women did not significantly influence the neurodevelopment of the child, as also verified in Regressions carried out (data not shown).”
line 306 - evidence to support this - do Haemoglobin levels in WHO for pregnant women not already reflect expanded plasma volume (they are lower than levels for non-pregnant women).
AUTHORS: Indeed, you are right in your discussion that the decrease in hemoglobin due to the expansion of plasma volume is already corrected at the cut-off points proposed by WHO for pregnant women. So, this part of the discussion sentence has been removed: “probably because plasma volume is expanding more rapidly than the red cell mass”.
line 331- evidence to support this statement?
AUTHORS: We have added it.
line 335 - please check grammar
AUTHORS: Done.
line 343 - I am not sure what is meant by the statement beginning " any adverse.....
AUTHORS: We have modified this sentence.
Reviewer 2 Report
The revised manuscript has resolved earlier concerns.
Table 4 Omission errors / posthoc analysis has two values: 0.045ab and 0.029ab. The latter presumably should be ac?
Author Response
Thank you for review our manuscript.
The revised manuscript has resolved earlier concerns.
Table 4 Omission errors / posthoc analysis has two values: 0.045ab and 0.029ab. The latter presumably should be ac?
Authors: The data are correct, the difference is between the groups a and b.
Reviewer 3 Report
I find this new version of the manuscript is ready for publication.
Author Response
Thanks for your review.